# Preeclampsia and Its Impact on Human Milk Activin A Concentration

**DOI:** 10.3390/nu15194296

**Published:** 2023-10-09

**Authors:** Alessandra Coscia, Lorenzo Riboldi, Elena Spada, Enrico Bertino, Stefano Sottemano, Ignazio Barbagallo, Giovanni Livolti, Fabio Galvano, Diego Gazzolo, Chiara Peila

**Affiliations:** 1Neonatal Unit, Department of Public Health and Pediatrics, University of Turin, 10124 Torino, Italy; alessandra.coscia@unito.it (A.C.); elenaspada.bios@gmail.com (E.S.); enrico.bertino@unito.it (E.B.); stefano.sottemano@unito.it (S.S.); chiara.peila@unito.it (C.P.); 2Department of Biological Chemistry, Medical Chemistry and Molecular Biology, University of Catania, 95123 Catania, Italy; ignazio.barbagallo@unict.it (I.B.); livolti@unict.it (G.L.); fabio.galvano@unict.it (F.G.); 3Neonatal Intensive Care Unit, Università degli Studi G. d’Annunzio Chieti e Pescara, 66100 Chieti, Italy; diego.gazzolo@unich.it

**Keywords:** human milk, breastfeeding, activin A, preeclampsia, newborn nutrition, neurobiomarker

## Abstract

Background: It is known that preeclampsia affects lactogenesis. However, data on the effects of this pathology on human milk neurobiomarker composition are not available. The aim of this study is to investigate the effects of this gestational pathology on activin A levels, a neurobiomarker known to play an important role in the development and protection of the central nervous system. Methods: The women recruited were divided in two different study groups: preeclamptic or normotensive women. All the human milk samples were collected using the same procedure. Activin A was quantified using an Enzyme-linked immunosorbent assay (ELISA) test. To investigate the effect of preeclampsia on the activin A concentration in the three lactation phases, a mixed linear model with a unistructural covariance structure, with the mother as the random effect, and fixed effects were performed. Results: Activin A was detected in all samples. There were no significant differences between preeclamptic and normotensive women. The only significant effect is related to the lactation phase: the difference between colostrum and mature milk (*p* < 0.01) was significant. In conclusion, these results allow us to affirm that breast milk’s beneficial properties are maintained even if preeclampsia occurs.

## 1. Introduction

Preeclampsia (PE) is a gestational hypertensive syndrome characterized by a complex disease with variable clinical manifestation [1]. In most cases, the symptoms occur after the 20th week of gestational age (GA); the earlier its onset, the more serious it is [2,3]. PE is a common pregnancy complication that occurs in about 2–8% of pregnancies globally. At the basis of PE, there is an early functional alteration of unknown origin to the development of placental-vascularization [3]. Among the advanced etiological hypotheses, an imbalance in the pro- and antiangiogenic factors, a reduced tolerance towards the child or the father, or insulin resistance can be found [4]. PE is one of the primary causes of fetal–maternal morbidity and mortality. In addition, infants born to preeclamptic mothers are at high risk for several disorders—not only endocrine, nutritional, and metabolic, but also cognitive due to neurodevelopmental impairments [5,6].

An aspect of interest is the possible effect that PE may have on the mammary gland, and therefore on breast milk composition. In fact, it is well known that human milk (HM) is a peculiar food owing unique properties and resulting in the ideal nourishment during the neonatal period for a growing infant [7,8,9,10]. HM is an individual species-specific biological “dynamic” system characterized by an extreme variability in its composition, with regard to both nutritional and bioactive components, changing according to the lactation phase and adapting its composition to different conditions such as GA, gestational pathologies, and/or maternal diet [7]. Mother’s own milk is always considered the first choice for nutrition of all neonates, including preterm newborns, thanks to the HM-specific biological active factors (i.e., hormones, immunoglobulins, lysozyme, lactoferrin, saccharides, nucleotides, and neurobiomarkers) that improved several neonatal outcomes both in the short- and the long-term [8,9,10]. Neurobiomarkers are important HM components, and between these, activin A can play a relevant role as a growth factor [11,12].

Activin A is a dimeric protein belonging to the transforming growth factor beta (TGF-beta) superfamily, and its receptors are widely distributed in the central nervous system (CNS) [13]. Studies in humans and the animal model showed that activin A can play a trophic and neuroprotective role on the CNS [13,14,15,16]. Several in vitro and in vivo studies have investigated the effect of activin A on the nervous system, showing that: (i) it acts as a nerve tissue survival factor [17], a neural differentiation inhibitor [18], and a mitogen factor [19]; (ii) is a powerful survival factor for neurogenic clonal cell lines, retinal neurons, and dopaminergic neurons of the midbrain [20]; (iii) promotes in vitro survival of hippocampal neurons from rat embryos [21]. In addition, activin A promotes the survival of specific populations of damaged neurons, and its expression is crucial for neuronal protection in case of brain injury [14,22]. In fact, it has been shown to protect the dopaminergic neurons of the midbrain from neurotoxic damage [23] and, in experiments on rats, to recover striatal neurons undergoing neurotoxicity [24]. It also performs, on the other hand, a biomarker function of damage, especially at the brain level. Therefore, it has also been suggested to use it as an early neonatal indicator of neurological insults caused, for example, by asphyxia and intraventricular hemorrhage after birth [24]. There is also evidence that the protective role of activin A is also extended to heart tissue. In particular, it has been reported that the protein can participate in a cascade of events, promoting tissue protection and regeneration in patients who have undergone ischemia/reperfusion injury [25,26].

During pregnancy, the predominant source of activin A is the placenta, which expresses both βA-mRNA and molecule receptors. High levels of activin A are present in the amniotic fluid, in which the concentration increases as pregnancy progresses, and in the celomatic fluid, which performs the function as a reservoir of activin A for the development of the fetus, although the molecule is also produced in fetal tissues [16]. Its levels rise in the maternal plasma during gestation, reaching maximum values at the end of the same; in particular, the serum level of activin A is higher in women who give birth with vaginal delivery than those who undergo a caesarean section, suggesting, therefore, an effect of the molecule on the mechanisms of childbirth (through the liberation of prostaglandins and oxytocin). In the case of PE, many studies agree that there is a significant correlation between blood and activin A levels [26]. Moreover, high levels of activin A in mid-trimester may be helpful in predicting which patients will develop preeclampsia during pregnancy. HM activin A is expressed by the mammary gland (βa-subunit and βa-mrna have been localized in ductal and lobular epithelial cells). It therefore appears to have at least two other important functions [12]: (i) it is necessary for the proper development of the mammary gland itself, acting as an autocrine and paracrine factor—if the gene encoding activin A is inactivated, the development of the breast is incomplete, and there is no milk production, the elongation of the ducts is not complete, and the morphogenesis of the secreting alveolar ducts is altered; (ii) it is likely to have a growth factor function on various tissues of the newborn (including the brain and heart tissues mentioned above) and an immune function. Activin A, in fact, increases the production of cytokines by mononucleate cells, regulates the development of T cells, and performs both pro-and anti-inflammatory actions.

The presence of HM activin A was evaluated for the first time by Luisi et al. [12] in women that delivered at term of GA. The results of this study show no differences in activin A levels for type of delivery, maternal age, or gestational age. There are also no differences between colostrum and transitional milk samples. The only significant difference lies in the molecule concentrations in mature milk, which are significantly lower than those of colostrum [12].

Although it is known that PE affects lactogenesis, literature data on the effects of this syndrome on the neurobiomarker composition and activin A of HM of the lactating mother are not available. Thus, the aim of this study is to integrate and to expand the available literature data by investigating the association between the composition of human milk and PE, considering the variations of this key biochemical marker during different lactation phases in mothers having delivered term and preterm infants.

## 2. Methods

### 2.1. Setting and Population

Mothers admitted into the study gave signed and informed consent. Newborns’ mothers were recruited after delivery at the Neonatal Unit of Department of Public Health and Pediatrics, University of Turin, Italy. Written informed consent was obtained from all participants, and approval from the local ethics committee was obtained.

The women recruited in the study were divided in the two different study groups: preeclamptic, according to the PE definition (artery blood pressure > 140/90 after 20 weeks of gestational age and proteinuria > 290 mg/L, possibly associated with headache, edema, scotomas, and epigastric pain [1]) or normotensive women. Exclusion Criteria: presence of diabetes mellitus, chorioamnionitis, all CNS pathologies or psychiatric syndromes; use of illicit drugs/alcohol during pregnancy; mastitis or continuous use of medication; newborn with congenital anomalies or infection.

The control group was formed at the same time and made up of normotensive mothers who met the same exclusion criteria. A minimum of 30 women for each group were recruited.

### 2.2. Collection of Human Milk Samples

According to standard criteria, we classify “colostrum” as the milk collected in the first three days after the delivery, “transition milk” as the milk collected from the 8th day to 14th day after the delivery, and “mature milk” as the milk collected after the 30th day [27]. All the breast milk samples were collected using the same procedure outlined below. Fresh milk samples were collected in the morning (between 9 a.m. and 12 a.m.) into sterile disposables. Milk was collected with standard extraction methods by emptying one breast completely by means of an electric breast pump (Medela Symphony, Baar, Switzerland). A minimum of 10 mL of milk was collected and immediately frozen at −80 °C before the analysis. Milk expression by the other breast was performed only in cases in which it is not possible to obtain 10 mL from a single breast.

### 2.3. Activin A Measurements

Activin A levels were determined using a specific ELISA test (ELH-ActivinA-1 Human Activin A ELISA) according to the manufacturer’s instructions (RayBiotech, Inc.; Peachtree Corners, GA, USA). Investigators who performed the laboratory tests were blind to storage modalities. The assay detection limit is 15.00 pg/mL, the intra-assay CV is ≤5.0%, and the inter-assay CV is ≤10%. The assay is specific for activin A, having been tested by the manufacturer for lack of cross-reactivity with other proteins of the activin family.

### 2.4. Statistical Methods

In the description of the sample, the categorical variables were presented as frequencies (percent), while the continuous variables were presented as mean (standard deviation) or median [interquartile range] (IQR) according to their distribution.

Birth weight was transformed into z-score using INeS charts as reference [28]. Newborns with a birth weight lower than the 10th or higher than the 90th percentile were classified as small for GA (SGA) or large for GA (LGA), respectively. The continuous variables were summarized as mean (standard deviation) or median [inter quartile range] according to their distribution; the categorical variables were summarized as absolute frequency (percent). To investigate the distribution by HM phase and pathology of activin A, a specific box plot was created. The comparison between the median of pathology and non-pathology by the HM phase was tested using a Kruskal–Wallis test. Then, activin A concentration was normalized with the more appropriate Box-Cox transformation. To investigate the effect of pathology in the activin A concentration in the 3 phases, a mixed linear model with unistructural covariance structure, with the mother as a random effect, and fixed effects were performed. The fixed effects were: HM phase, pathology, smoke, type of delivery, GA and mother age (continuous), and the interaction between phase hm×pathology.

## *3.* Results

A total of 82 mothers were recruited for our study, divided as follows: 36 in the PE group and 49 in the normotensive group. Table 1 reports the basal characteristic of mothers and newborns included in this study. Regarding the maternal characteristics, in both groups, the median age was quite similar (median age: 35 in the PE group and 33.5 in the normotensive group) as well as the fraction of primigravida (64% in the PE group and 63% in the normotensive group). Moreover, the percentage of Caesarean section was high in the two groups (28% in the PE group and 25% in the normotensive group).

Concerning the neonatal characteristics, as expected, the women with PE had a higher fraction of IUGR (41% in the PE group and 4.4% in the normotensive group) and SGA (50% in the PE group and 13% in the normotensive group) newborns.

### 3.1. Characteristics of the Human Milk Samples

A total of 158 HM samples were collected. In particular, concerning the normotensive group, a total of 79 samples were collected, of which 30 were colostrum, 24 of transitional milk, and 25 of mature milk. In the PE group, a total of 79 samples were collected, of which 30 were colostrum, 27 of transitional milk, and 22 of mature milk.

### 3.2. Activin A Concentrations

Activin A was detected in all samples and in all types of HM, regardless of the lactation phase, gestational pathologies, and the GA at childbirth.

In the normotensive group, the activin A median concentration was, respectively: 232.47 pg/mL [IQR 96.13–771.46] in colostrum samples, 122.47 pg/mL [IQR 74.80–254.80] in transitional milk samples, 147.46 pg/mL [IQR 82.80–260.80] in mature milk samples, 142.13 pg/mL [IQR 71.46–280.63] in HM samples of women who delivered at term of GA, and 232.46 pg/mL [IQR 132.96–744.46] in HM samples of women who delivered preterm of GA. In the PE group, the activin A median concentration was, respectively, 553.80 pg/mL [IQR 340.13–751.46] in colostrum samples, 238.80 pg/mL [IQR 152.80–428.13] in transitional milk samples, 108.13 pg/mL [IQR37.46–274.80] in mature milk samples, 703.46 pg/mL [IQR 452.13–1141.58] in HM samples of women who delivered at term of GA, and 475.46 pg/mL [IQR 206.13–577.46] in HM samples of women who delivered preterm of GA.

Figure 1 shows the boxplot of activin A distribution by the HM phase and groups. The variability in HM phase 1 (colostrum) is higher than in phase 2 (transitional milk) and 3 (mature milk). Activin A concentration was significantly reduced between PE and the normotensive group in colostrum (Kruskal–Wallis test *p* = 0.05) and transitional milk (Kruskal–Wallis test *p* = 0.02), though not in mature milk (Kruskal–Wallis test *p* = 0.26). Such differences disappear when applying a mixed linear model (λ = 0 Box-Cox transformation to normalized activin A distribution), where no significant effect of pathology and phase hm×pathology resulted. The only significant effect was related to the HM phase, in particular, the significant difference between colostrum and mature milk (*p* < 0.01).

## 4. Discussion

Among the maternal organs affected by PE, there are the mammary glands. Throughout the rest of the body, even at this level, there could be changes in the endothelium and blood vessels; these would lead to a reduction in the development of the gland and changes in the mechanisms of the production of milk [29]. It is also known that the children of preeclamptic mothers, exposed to intrauterine stress, may have special nutritional needs in addition a greater risk of complications [2,30]. In view of these considerations, it is interesting to evaluate the potential differences in composition of HM between PE women and normotensive women in the different lactation phases. 

Our study is the first that provides data on the association between PE and HM activin A levels. Our results show the absence of significant differences between the two women’s groups when a mixed linear model was performed.

Considering the importance of HM in newborn nutrition, previous studies have focused their attention on a PE lactating mother. Data shows an alteration in the levels of several components: macronutrients (i.e., proteins, carbohydrates, lipids, and energy metabolites) and pro- and anti-inflammatory cytokines, oxidative stress markers, and antioxidant molecules [31,32,33,34,35,36]. Specifically, two previous studies evaluated the effects of PE on two similar panels of cytokines of HM, and both considered only the variations between colostrum to mature milk [33,34]. Data of both studies demonstrated that PE modifies the inflammatory cytokine levels in HM as well as the cytokine profile, and these modifications depended on the lactation stage. The first study showed that pro-inflammatory cytokines (IL-1β, sIL-2R, IL-6, IL-8, and TNF-α) in HM displays biological differences in different periods of lactation, that is, higher cytokine levels in the colostrum versus mature milk following normal pregnancy. In PE, high cytokine levels perdure in mature milk, but all cytokines’ concentrations were not significantly different in the PE versus the control group in colostrum. However, IL-8 and TNF- α levels were higher in the PE group versus control in mature milk [33]. The data of the other study showed that in the colostrum of the PE group, IL-1b and IL-6 levels increased and IL-12 levels decreased, whereas in the mature milk, IL-6 and IL-8 levels decreased more than those of the control. Regarding the differences during the lactation period, in the control group, the levels of IL-8, IL-10, and IL-12 were lower in mature milk than in colostrum, whereas the IL-6 concentration was higher in mature milk. In opposition, many cytokine levels in PE were stable and showed no differences between colostrum and mature milk; only IL-1β and IL-8 decreased during the postpartum period [34]. The levels of most cytokines did not decrease as lactation progressed, which may reflect the persistence of the systemic inflammatory response or a change in the immune system of the mammary gland in women with PE [33,34]. PE is a systemic inflammatory disease, so it is interesting to speculate whether the inflammatory response also occurs in the mammary gland, leading to increased levels of inflammatory cytokines in human milk. Moreover, cytokine production in the mammary gland is an active process, so the reduction of IL-12 in the milk of mothers with PE may represent a defense mechanism for a neonate exposed to a chronic inflammatory condition during fetal life [37,38]. Another interesting finding in the PE group was the lower concentrations of IL-6 and IL-8 in mature milk. These cytokines are produced in the mammary gland, so this decrease during the progression of lactation may reflect persistent adaptations in the mother’s body to protect the newborn. However, IL-8 plays a trophic role in the intestinal mucosa of the infant, so decreased values can mean less protection for the gastrointestinal tract [37]. IL-1β increased levels in colostrum can be beneficial to the newborn because it appears to be involved in human milk defense mechanisms, including the production of IgA and other cytokines [39].

Regarding neurotrophic factors, Dangat et al. examined the levels of brain-derived neurotrophic factor (BDNF) and nerve growth factor (NGF) [31,40]. At first, they evaluated the neurotrophin levels only in colostrum and observed that NGF levels were similar, whereas BDNF levels were higher in the PE group as compared to controls [31]. During the second time, they extended the evaluation of these agents through the other phases of lactation, and they found that the NGF concentrations at 1.5 and 3.5 months and BDNF levels at 1.5 months were lower in the PE group as compared to the control group [40]. BDNF and NGF are known to play a critical role in the development and maintenance of the nervous system. The significant quantitative differences in this neurotrophin at several time points during lactation probably indicate that milk programming by the mother’s breast is altered by preeclampsia [40]. It is likely that these changes are most likely adaptive changes of the mother; some of these changes are not normalized, even up to 6 months [40].

In particular, activin A probably acts in HM as a growth factor.

Our data show, like the previous study of Luisi et al. [12], differences in activin A concentration in the different lactation phases, with a significant decrease in levels from colostrum to mature milk, and also in mothers having delivered preterm and term of GA. In addition, this current study also confirms the presence of activin A in the HM of woman that delivered preterm, as demonstrated only in a single previous study [11].

The absence of differences in HM activin A composition is an important finding; it can be said that the beneficial properties of milk are maintained even in the event of the onset of PE. 

Future studies are needed to confirm the present findings and to obtain a more comprehensive evaluation of the effects of this important pathology on HM. 

## 5. Conclusions

Given the importance of nutrition of newborns with breast milk, many studies have investigated the characteristics of HM of women with gestational pathologies. Some of them have focused on women with gestational hypertension and/or preeclampsia, comparing the composition of their milk with that of normotensive mothers. The main observation on the data reported in the literature is that PE altered the composition of HM. The effect of PE on HM is present not only in the first day post-partum, but continues throughout all the lactating phases. In some cases, the modification in concentration of biological factors of HM may appear as a benefit for newborns, but the variation of other components is a disadvantage for the development of these babies [41]. Currently, however, there is still little known on the influence of this pathology on the composition of milk. The objective of our study was to fill part of this gap in the literature. Our study is the first that extensively evaluates the concentration of these molecules in the milk of women with PE, regardless of the gestational age at delivery and taking into account all stages of lactation. In conclusion, our study shows that there is not a significant difference in HM activin A composition from PE and normotensive women, so we can hypothesize that the biological value of human milk associated with activin A content is maintained.

Further studies are needed to confirm our findings and to identify the potential correlations between the composition of HM and the different gestational hypertension pathologies.

## Figures and Tables

**Figure 1 nutrients-15-04296-f001:**
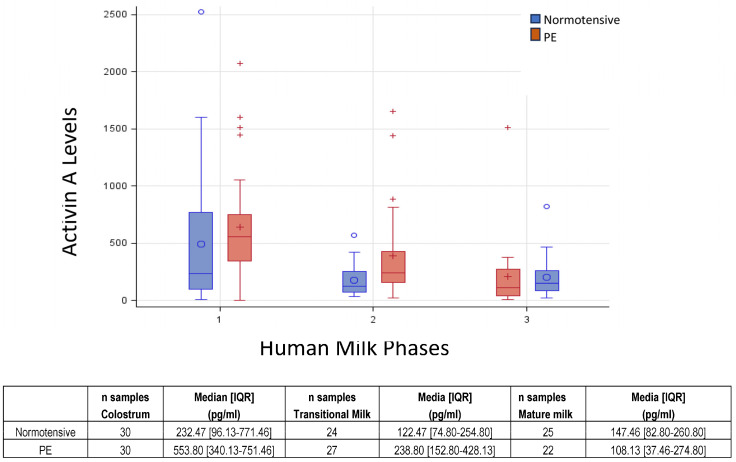
Boxplot of activin A by HM phase and pathology. Median and [IQR] were reported for each HM phase: 1—colostrum; 2—transitional milk; 3—mature milk.

**Table 1 nutrients-15-04296-t001:** Basal characteristics of mothers and newborns.

		Normotensive *N* = 46	Preeclamptic *N* = 39
** *Maternal characteristics* **			
Age (years)	median [IQR]	33.5 [31–37]	35 [31–38]
Italian	*n* (%)	35 (76.1)	31 (79.5)
Caesarian Section	*n* (%)	25 (54.4)	28 (71.8)
Weight gain (kg)	mean (SD)	10.9 (4.75)	10.4 (5.65)
Primigravida	*n* (%)	29 (63.0)	25 (64.0)
Smoker	*n* (%)	6 (13.0)	2 (5.1)
** *Newborn characteristics* **			
Singleton	*n* (%)	38 (82.6)	36 (92.3)
IUGR	*n* (%)	2 (4.4)	16 (41.0)
GA (weeks)	median [IQR]	37 [31;39]	32 [29–35]
Girls	*n* (%)	19 (41.3)	19 (48.7)
Birth weight (g)	mean (SD)	2345 (1028)	1542 (720)
Birth weight (z-score)	mean (SD	−0.21 (0.934)	−1.16 (0.810)
SGA	*n* (%)	6 (13.0)	19 (50.0)
LGA	*n* (%)	2 (4.4)	0 (0.0)

## Data Availability

The data presented in this study are available on request from the corresponding author.

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
