# Peer review of "Preeclampsia and Its Impact on Human Milk Activin A Concentration"

_nutrients, 2023, doi:10.3390/nu15194296_

Round 1
Reviewer 1 Report
Although there are some ineteresting results presented in this study, a general revision has to be done.
First, the authors should clearly explain, why they choose just Activin A.
Second, many intersting information about activin A is given in the discussion, but that is the wrong place. This part has to be improved.
Other parts have to be improved too:
In particular:
line 43, 44: reference missing, general statement
line 49, 50, 51: interesting hypothesis...reference???
line 69: methods
line 79: tense, this sentence seems to be lonely...
line 94: was collected (?)
line 129: please give the data here
table one:
- why was the birth weight 2345 g in SSW 37? This is low.
- Also I am wondering, why the median GA was 37 (!!) in the normotensive group?
line 279, 283, 284 (only 3 examples, please revise the discussion!!!): These facts should be explained in the introduction!
line 294, 295 should be moved to the introduction!!
line 305 - 310: leave the emotions out of it!
"thanks to these data" ???
line 312, 313: "It will be also important to analyze the effect of different drugs of PE on HM and their potential interaction on the different biological components:" -> delete. This is a different task.
line 318, 319: delete or give a reference and explain why this is a CONCLUSION!
Limitations of the study: I am not very familiar with statistics but wondering if it is possible to make ANY conclusions from a sample number of 30 (!). I would not publish data from 30 individuals and 30 controls.
Author Response
Dear Sir / Madam,
enclosed please find the revised version of the MS entitled “Preeclampsia and their impact on human milk Activin A concentration” by Coscia A.; Riboldi L. et al. We want to thank you and the reviewer for their kind comments and useful suggestions that have been taken into account in the present revised form. In particular:
First, the authors should clearly explain why they choose just Activin A. Second, many interesting information about activin A is given in the discussion, but that is the wrong place. This part has to be improved.
We have carried out an extensive review of both the introduction and the discussion on the basis of the suggestions made.
In particular:
- line 43, 44: reference missing, general statement
We added references as suggested.
- line 49, 50, 51: interesting hypothesis...reference???
The sentence has been deleted
- line 69: methods
Typing errors have been corrected in the text as suggested.
- line 79: tense, this sentence seems to be lonely...
The sentence has been moved to the text as suggested
- line 94: was collected
The text has been changed as requested.
- line 129: please give the data here
The data has been added into the text as suggested
Table one:
- why was the birth weight 2345 g in SSW 37? This is low.
- Also I am wondering, why the median GA was 37 (!!) in the normotensive group?
Medians of birthweight and GA in our populations are low because mothers were enrolled regardless of GA at childbirth matching inclusion/exclusion criteria since we started the protocol.
- line 279, 283, 284 (only 3 examples, please revise the discussion!!!): These facts should be explained in the introduction!
- line 294, 295 should be moved to the introduction!!
We moved the paragraphs indicated in the introduction going to rewrite this section and also the discussion section.
- line 305 - 310: leave the emotions out of it! "thanks to these data"
Text has been changed as recommended
- line 312, 313: "It will be also important to analyze the effect of different drugs of PE on HM and their potential interaction on the different biological components:" -> delete. This is a different task.
- line 318, 319: delete or give a reference and explain why this is a CONCLUSION!
Sentences were deleted as suggested
Limitations of the study: I am not very familiar with statistics but wondering if it is possible to make ANY conclusions from a sample number of 30 (!). I would not publish data from 30 individuals and 30 controls.
Our data collection refers to the analysed human milk samples, corresponding to a total of 158 HM samples (79 concerning the Normotensive group, and 79 samples in the PE group)
We trust that now, the MS in the present revised version will meet the criteria for publication in the Nutrients.
Best regards
Reviewer 2 Report
The current article by Coscia et al investigates presence of Activin-A in the milk of pre-eclampsiatic (PE) and normotensive (NE) mothers at the different stages of lactation. The authors have concluded that there are no significant differences between preeclamptic and Normotensive women and breast milk beneficial properties are maintained even if preeclampsia occurs.
Overall, the study is well designed, provides primary and brief information confirming the previous report. However, I have some major concerns,
1) Despite Activin A being the focus of paper, and its presence on other types of samples and its role has been discussed extensively, the current manuscript provides very brief literature review on the topic.
2) The major conclusions drawn in this study by authors are that there were no significant differences between Preeclamptic and Normotensive women and the significant effect is related to the lactation phase. As per the table provided in figure 1, normotensive and PE group at least colostrum and transitional stage of lactation show significant differences.
3) The figure 1 isn’t self-explanatory. The sequence of the subgroups (PE and Normotensive) the last group of comparison (Mature milk) is incorrect. Can authors add statistical analysis for
a) comparisons between PE Vs Normotensive at different lactation phase
b) comparison between different lactation phase per group
4) In order to derive the conclusion that breast milk beneficial properties are maintained even if preeclampsia occurs, in addition to looking at the levels of Activin-A, it will be interesting and equally important to look at the levels of the other nutritional factors and cytokines and in the milk of the cohort include in this study.
5) The PE group shows a trend where the high level of Activin-A reduces and "normalizes" as the lactation phase progresses towards mature milk stage. Is there any rational behind these changes and whether it could it be affecting the growth of the baby?
Author Response
Dear Sir / Madam,
enclosed please find the revised version of the MS entitled “Preeclampsia and their impact on human milk Activin A concentration” by Coscia A.; Riboldi L. et al. We want to thank you and the reviewer for their kind comments and useful suggestions that have been taken into account in the present revised form. In particular:
1) Despite Activin A being the focus of paper, and its presence on other types of samples and its role has been discussed extensively, the current manuscript provides very brief literature review on the topic.
We have carried out an extensive review of both the introduction and the discussion on the basis of the suggestions made.
2) The major conclusions drawn in this study by authors are that there were no significant differences between Preeclamptic and Normotensive women and the significant effect is related to the lactation phase. As per the table provided in figure 1, normotensive and PE group at least colostrum and transitional stage of lactation show significant differences.
As the reviewer noted, the raw values differ in stages 1 and 2. This difference disappears when applying the mixed linear model due to the random effect of the subjects and not the pathology.
3) The figure 1 isn’t self-explanatory. The sequence of the subgroups (PE and Normotensive) the last group of comparison (Mature milk) is incorrect. Can authors add statistical analysis for
- a)comparisons between PE Vs Normotensive at different lactation phase
- b)comparison between different lactation phase per group
The analyses have been revised adding comparisons between groups.
4) In order to derive the conclusion that breast milk beneficial properties are maintained even if preeclampsia occurs, in addition to looking at the levels of Activin-A, it will be interesting and equally important to look at the levels of the other nutritional factors and cytokines and in the milk of the cohort include in this study.
Thank you for your suggestion. We have modified the text to be more precise in our statements in the conclusions. Further future studies could be developed to assess the levels of other nutritional factors in the breast milk
5) The PE group shows a trend where the high level of Activin-A reduces and "normalizes" as the lactation phase progresses towards mature milk stage. Is there any rational behind these changes and whether it could it be affecting the growth of the baby?
Thank you for your suggestion. Further future studies could be developed to better understand these changes and the impact on the growth of babies.
We trust that now, the MS in the present revised version will meet the criteria for publication in the Nutrients.
Best regards
Round 2
Reviewer 1 Report
Past and present tense are still mixed up.
It would be nice to have some more information about the statistics to asses the significance of these methods.
Reviewer 2 Report
The authors have addressed the all the comments satisfactorily and have made the necessary changes in manuscript.
The current version of manuscript can be accepted in the present form for publication.